# Probabilistic assessment of postfire debris-flow inundation in response to forecast rainfall

Alexander B. Prescott[1], Luke A. McGuire[1], Kwang-Sung Jun[2], Katherine R. Barnhart[3], Nina S. Oakley[4]

[1]Department of Geosciences, The University of Arizona, Tucson, AZ, USA
[2]Department of Computer Science, The University of Arizona, Tucson, AZ, USA
[3]U.S. Geological Survey, Denver, CO, USA
[4]California Geological Survey Burned Watershed Geohazards Program, Sacramento, CA, USA

*Correspondence to*: Alexander B. Prescott (alexprescott@arizona.edu)

**Abstract.** Communities downstream from burned steeplands face increases in debris-flow hazards due to fire effects on soil and vegetation. Rapid postfire hazard assessments have traditionally focused on quantifying spatial variations in debris-flow likelihood and volume in response to design rainstorms. However, a methodology that provides estimates of debris-flow inundation downstream from burned areas based on forecast rainfall would provide decision-makers with information that directly addresses the potential for downstream impacts. We introduce a framework that integrates a 24-hour lead-time ensemble precipitation forecast with debris-flow likelihood, volume, and runout models to produce probabilistic maps of debris-flow inundation. We applied this framework to simulate debris-flow inundation associated with the 9 January 2018 debris-flow event in Montecito, California, USA. When the observed debris-flow volumes were used to drive the probabilistic forecast model, analysis of the simulated inundation probabilities demonstrates that the model is both reliable and sharp. In the fully predictive model, however, in which debris-flow likelihood and volume were computed from the atmospheric model ensemble's predictions of peak fifteen-minute rainfall intensity, $I_{15}$, the model generally under-forecasted the inundation area. The observed peak $I_{15}$ lies in the upper tail of the atmospheric model ensemble spread, thus a large fraction of ensemble members forecast lower $I_{15}$ than observed. Using these $I_{15}$ values as input to the inundation model resulted in lower than observed flow volumes which translated into under-forecasting of the inundation area. Even so, approximately 94% of the observed inundated area was forecast to have an inundation probability greater than 1%, demonstrating that the observed extent of inundation was generally captured within the range of outcomes predicted by the model. Sensitivity analyses indicate that debris-flow volume and two parameters associated with debris-flow mobility exert significant influence on inundation predictions, but reducing uncertainty in postfire debris-flow volume predictions will have the largest impact on reducing inundation outcome uncertainty. This study represents a first step toward a near-real time hazard assessment product that includes probabilistic estimates of debris-flow inundation and provides guidance for future improvements to this and similar model frameworks by identifying key sources of uncertainty.

## 1 Introduction

Debris flows threaten life and property in mountainous areas worldwide (Dowling and Santi, 2014). In the first few years following fire, debris-flow hazards are greater relative to nearby unburned areas due to reductions in soil infiltration capacity (Ebel, 2020; Larsen et al., 2009; McGuire and Youberg, 2020), decreased vegetation cover (Cerdà and Doerr, 2005; Hoch et al., 2021; Stoof et al., 2012), increases in sediment availability (Nyman et al., 2013), and reduced thresholds for sediment entrainment (Moody et al., 2005). Postfire debris flows typically initiate when short-duration, high-intensity rainfall generates runoff that rapidly entrains sediment from hillslopes and channels (DeGraff et al., 2015; Gabet and Bookter, 2008; Kean et al., 2011; McGuire et al., 2017). Fire-related debris flows pose hazards globally, including in Europe (Conedera et al., 2003; Diakakis et al., 2023; Esposito et al.,

2023; García-Ruiz et al., 2013; Lourenço et al., 2012), Asia (Jin et al., 2022; Lee et al., 2022; Touge et al., 2023), western North America (e.g., Jordan, 2015; Jordan and Covert, 2009; Kean et al., 2019), and Australia (Nyman et al., 2011). Due to the potential for widespread (i.e., within hundreds of different watersheds) increases in debris-flow susceptibility following fire, it is critical to rapidly identify downstream areas most threatened by debris-flow runout and to quantify the uncertainty associated with their identification.

Empirical models that assess debris-flow likelihood and volume at the drainage basin scale have been developed for the western United States in the 1-2 years immediately following wildfire (Gartner et al., 2014; Staley et al., 2017). These models can be used to produce rapid hazard assessments in response to design rainstorms with spatially uniform intensity (e.g., Staley et al., 2016). Past studies demonstrate that, in addition to factors related to topography and soil burn severity, rainfall intensity over a 15-minute duration ($I_{15}$) controls the likelihood of debris-flow initiation within a basin (Staley et al., 2017) as well as debris-flow volume (Gartner et al., 2014). Forecasts of debris-flow volume are highly uncertain, especially when applied in settings not represented within their training datasets (Gorr et al., 2023; Wall et al., 2023), and may exceed a factor of 10 (Gartner et al., 2014). Debris-flow likelihood and volume both increase with $I_{15}$, which indicates that variations in rainfall intensity, even over small spatial (i.e., a low-order basin) and temporal (15-minute) scales, will play an important role in determining the likelihood and spatial extent of debris-flow impacts. Assessing the downstream impact of debris flows, however, requires information about their runout and inundation extent.

Rapid postfire hazard assessments (e.g., Staley et al., 2016) do not currently provide information about downstream impacts, although recent debris-flow events (Kean et al., 2019) and surveys of the postfire hazards emergency management community highlight the need for such a product (Barnhart et al., 2023; Gourley et al., 2020). The probability that a downstream area will be impacted by a debris flow depends, in part, on the likelihood of a debris flow initiating in a basin upstream, the size (volume) of the debris flow, and the movement of the flow as driven by topography and flow dynamics. Several debris-flow inundation models have been applied to simulate a recent series of postfire debris flows following the 2017 Thomas Fire in southern California (Barnhart et al., 2021; Gibson et al., 2022; Gorr et al., 2022). The models used in these studies vary in their representation of debris-flow physics, from those that account for multiphase flow dynamics and pore-pressure feedbacks (George and Iverson, 2014; Iverson and George, 2014) to those that use a semi-empirical approach to route flow across the landscape (Gorr et al., 2022). All the inundation models require debris-flow volume as an input, although some models require an inflow hydrograph (Barnhart et al., 2021), whereas others require volume be specified as a single number at each initiation point representing the total volume mobilized (Gorr et al., 2022). The model proposed by Gorr et al. (2022), the Progressive Debris-Flow routing and inundation model (ProDF), performed similarly to other inundation models but required less run time for equivalent simulations. This makes it a promising tool for evaluating runout for a large number of forecast precipitation scenarios in a rapid hazard assessment framework.

Existing debris-flow likelihood, volume, and runout models therefore provide the necessary components to create a framework for postfire hazard assessments that includes probabilistic estimates of inundation area in response to a forecast or design rainstorm, but such a framework has yet to be developed and explored. Probabilistic frameworks for predicting debris-flow runout have been explored in unburned settings, although runout models have not been directly linked with others designed to predict debris-flow likelihood and volume (Aaron et al., 2022; Sun et al., 2021). As a result, fundamental questions remain regarding the propagation of uncertainty through various model components (i.e., from rainfall to flow volume to runout) as well as the benefits and limitations of such an approach at forecast lead times ($\geq$ 24 hours) needed for decision-making (e.g., evacuation, allocating resources).

A postfire debris-flow inundation hazard assessment should reflect uncertainty in forecast inundated area (Barnhart et al., 2023), and past work identifies at least three ways through which substantial uncertainty is likely to arise: (1) forecast peak $I_{15}$ needed as input for debris-flow likelihood and volume models (Oakley et al., 2023); (2) simulated debris-flow volume given a peak $I_{15}$ (Gartner et al., 2014); and (3) flow mobility parameters needed to drive a debris-flow runout model (Aaron et al., 2019). Precipitation forecasts in the weeks to hours ahead of an event include considerable uncertainty regarding short-duration and high-

intensity rainfall rates (Oakley et al., 2023). Assuming a perfect rainfall forecast, debris-flow volume models have order-of-magnitude uncertainty around a given prediction (Gartner et al., 2014). Lastly, debris-flow runout model parameters that influence flow mobility require calibration, a process subject to observation biases, model assumptions, and subjective user decisions (Aaron et al., 2019). Runout model parameter uncertainty can be considerable, particularly in areas without data from prior events to calibrate against (Zeng et al., 2023), resulting in poor predictive performance of debris-flow runout (e.g., Gorr et al., 2023). Because

prior studies have found that debris-flow runout is sensitive to flow volume (e.g., Barnhart et al., 2021; Gorr et al., 2022), and given that uncertainty in rainfall intensity propagates forward into debris-flow volume predictions, we propose a framework for generating probabilistic debris-flow inundation maps that links atmospheric modeling with debris-flow models.

    The main objectives of this work were to (1) develop an integrative atmosphere-debris-flow model framework to generate a spatially distributed forecast of inundation probability, (2) apply the proposed framework to assess debris-flow inundation

downstream from burned basins using an atmospheric model ensemble designed to mimic a 24-hour lead time forecast, and (3) quantify the relative importance of key input parameters using global sensitivity analyses. A probabilistic inundation map is the final product of our model framework. In practice, this product is one that could be used to improve situational awareness for decision-makers. This study is a first step toward the development of a near-real time framework for probabilistic assessments of debris-flow inundation downstream from recently burned areas and provide guidance for future work aimed at further quantifying

and reducing uncertainty.

## 2 Study area

    We focused our study on a portion of the 2017 Thomas Fire near the community of Montecito, California, USA, that was impacted by postfire debris flows in January 2018, causing 23 fatalities and substantial economic losses (Kean et al., 2019; Lancaster et al., 2021) (Fig. 1). The Thomas Fire burned more than 1100 $km^2$, including a series of steep basins in the Santa Ynez Mountains

upstream from Montecito. The fire ignited in December 2017 and was not yet contained when a weak atmospheric river with an embedded narrow cold frontal rainband (NCFR) impacted the area on 9 January 2018 (Oakley et al., 2018). As the NCFR propagated over the burned basins upstream from Montecito, it produced rainfall with peak $I_{15}$ between 78 mm h$^{-1}$ and 105 mm h$^{-1}$ (Kean et al., 2019). Runoff entrained sediment from burned hillslopes and channels (Alessio et al., 2021; Kean et al., 2019; Morell et al., 2021), producing debris flows that traveled several kilometers down the alluvial fan (Lancaster et al., 2021). We focused on

six of these debris-flow-producing basins where approximately 679,000 $m^3$ of sediment was mobilized and debris-flow inundation extent (more than 2,600,000 $m^2$) was mapped shortly following the event (Kean et al., 2019). Debris flows in the Montecito Creek Basin were sourced from two upstream burned basins, while the remaining four creeks had a single upstream source. All simulations were run using topography from a 5-meter resolution digital elevation model derived from airborne lidar collected before the event (Fig. 1).

## 3 Methods

### 3.1. Overview of model framework

We coupled rainfall output from an atmospheric model ensemble with debris-flow likelihood, volume, and runout models to generate a probabilistic forecast of postfire debris-flow inundation downstream from the six debris-flow producing basins (Fig. 2). We used a 100-member atmospheric model ensemble representing a 24-hour lead-time forecast of the 9 January 2018 precipitation event (Oakley et al., 2023). For each ensemble member, we computed basin-averaged values of peak 15-minute rainfall intensity ($I_{15}$) and used this as input into debris-flow likelihood (Staley et al., 2017) and volume (Gartner et al., 2014) models to predict (1) whether each basin would produce a debris flow as well as (2) the volume of sediment a debris flow, if initiated, would mobilize. We then used the ProDF debris-flow runout model (Gorr et al., 2022) to estimate downstream inundation extent and peak flow depths. In this step, we incorporated uncertainty in debris-flow volume for a given $I_{15}$ as well as uncertainty in ProDF flow mobility parameter values into the forecast by utilizing Monte Carlo sampling methods. Finally, we produced a map of spatially variable forecast probabilities of inundation by averaging the inundated area results from each ProDF simulation.

### 3.2. Atmospheric model ensemble design

The 24-hour lead time, 100-member ensemble rainfall forecast for the 9 January 2018 event (Oakley et al., 2023) was generated using the Weather Research and Forecast (WRF) atmospheric model Version 4.3 (Skamarock et al., 2021). The ensemble produced a distribution of precipitation rates that reflects forecast uncertainty (Fig. 1c). Output consisted of spatially variable rainfall depths in 5-minute intervals with 1-km horizontal resolution across the study area. We spatially averaged rainfall intensities over each of the six basins (ranging from 0.45 to 8.94 km$^2$; Kean et al., 2019) and used a 15-minute moving window to calculate the peak $I_{15}$ for each basin for every ensemble member. In this way, variability in the timing, location, and spatial structure of forecast precipitation translates into variability in the $I_{15}$ subsequently used to predict debris-flow likelihood, volume, and runout.

### 3.3. Debris-flow likelihood and volume models

We used empirical models for postfire debris-flow likelihood (Staley et al., 2017) and volume (Gartner et al., 2014) to determine if a basin would produce a debris flow and how large it would be. The Staley et al. (2017) M1 model determines debris-flow likelihood, whereas the Gartner et al. (2014) emergency assessment volume (EAV) model predicts debris-flow volume. These models use basin-averaged metrics related to topography, soil properties, soil burn severity, and peak $I_{15}$ as input. In this study, all input parameters for the M1 and EAV models were fixed for each basin with the exception of $I_{15}$.

Using the basin-averaged $I_{15}$ from each WRF ensemble member, we computed debris-flow likelihood for each of the six basins using the M1 model. If debris-flow likelihood was less than 0.5, we assumed that a debris flow would not initiate. If likelihood was greater than 0.5, we assumed a debris flow would initiate and determined its volume using the EAV model. We defined a log-uniform distribution centered on the EAV-predicted volume with an order-of-magnitude envelope above and below the predicted volume (Fig. S1). This range of support is consistent with the prediction uncertainty of the EAV model, as well as similar models (Gartner et al., 2014). For a given ProDF simulation, we drew input volumes from these distributions by sampling from a log-uniform random variable over the range [0.1, 10.0] and multiplying the six EAV-predicted volumes (one for each of the six basins) by this scalar.

**3.4 Debris-flow inundation model**

We used ProDF to simulate debris-flow runout and inundation (Gorr et al., 2022). The model requires two flow mobility parameters and input debris-flow volumes at user-defined flow starting points, from which flow is iteratively routed downstream. The two flow mobility parameters, $\chi$ [$s^{-1}$ $m^{-0.5}$] and $\tau_y$ [Pa], control the flow depth and the minimal basal shear stress that permits flow motion, respectively.

It is common to calibrate models by choosing a single set of "best" parameters based on some objective function optimization tied

to an observation dataset and to then use this optimal parameter set in forward model applications (e.g., Pirulli, 2010). A limitation of this methodology is that it precludes exploration of a wider range of equifinal possible parameter sets, and different observation datasets may lead to different choices about which parameters are optimal. Instead, we used a statistical inference procedure similar to that of Aaron et al. (2019) to define a joint posterior distribution over the flow mobility parameters that uses the similarity index (Gorr et al., 2022; Heiser et al., 2017) as the objective of the maximum likelihood estimator function for nonlinear systems (Hill

and Tiedeman, 2007). The similarity index varies from negative one to one, with one indicating a perfect match between the simulated and observed area inundated. We generated samples from this distribution using the *emcee* Python implementation (Foreman-Mackey et al., 2013) of the Markov Chain Monte Carlo (MCMC) affine invariant ensemble sampler (Goodman and Weare, 2010). This method is advantageous as we were able to sample uniformly from the MCMC output to gather flow mobility parameters for ProDF simulations. We calibrated the posterior distribution to the Oak, San Ysidro, Buena Vista, and Romero Creek

Basins and reserved Montecito Creek to test the calibrated distribution. Additional details on the ProDF calibration are given in Supporting Text S1.

**3.5. Inundation forecast**

We generated a forecast of inundation probability by averaging together many individual ProDF simulations that were run with input parameters drawn repeatedly and independently from the calibrated distributions (Fig. 2). We ran 50 simulations for every

WRF ensemble member's prediction of peak $I_{15}$ (i.e., a total of 5,000 ProDF simulations). Debris-flow volumes for all six basins were drawn from the log-uniform distributions defined in Section 3.3, and the flow mobility parameters ($\chi$, $\tau_y$) were drawn from the calibrated joint posterior distribution defined in Section 3.4. Every simulation produced a map of peak debris-flow depth. Depth maps from all simulations were converted to binary maps of inundation presence using a threshold depth of 10 cm (e.g., Gorr et al., 2022). Averaging the binary inundation maps together with equal weights produced a map with values between zero and one

representing the fraction of simulations that inundated each grid cell, which we interpret as an inundation probability.

To investigate the role of input debris-flow volume on the joint distribution of forecasts and observations (described further in Section 3.6), we also generated probabilistic inundation maps for two additional scenarios, referred to as scenarios A and B. In scenario A, we use observed debris-flow volumes as input for all simulations. In this scenario, we minimize uncertainty in debris-flow volume, so we expect model performance to improve relative to the forecast. In scenario B, we assigned a debris-flow volume

to each basin using the EAV prediction with the observed peak $I_{15}$ as the input (Kean et al., 2019). This scenario also does not utilize any data from the atmospheric model ensemble. The peak $I_{15}$ at each debris-flow initiation point was computed with inverse distance weighting of the observed rainfall rates at the KTYD and Doulton Tunnel (DT) rain gauges (78 and 105 mm $h^{-1}$, respectively; Kean et al., 2019) (Fig. 1c). Evaluating model performance when debris-flow volume (scenario A) or rainfall (scenario B) are known is useful for identifying the source of any observed over or underestimation of inundated area in the

forecast.

**3.6 Comparing simulated and observed inundation**

Debris-flow inundation model results are commonly assessed one simulation at a time by optimizing an objective function of the mapped debris-flow deposits and simulated inundation zones (e.g., Barnhart et al., 2021; Gibson et al., 2022; Gorr et al., 2022). Probabilistic forecasts cannot be directly evaluated with a similar binary classification and optimization procedure. Instead, we classified grid cells as inundated or not using a threshold probability $p_t$ to explore the extent to which the observation was contained within the range of inundation scenarios represented by the ensemble forecast. If a given cell's forecast probability of inundation $p$ satisfied $p \geq p_t$, it was classified as inundated, and otherwise it was not. For values of $p_t$ between zero and one (discretized every 0.01), we classified each cell in the domain and then computed the similarity index.

Probabilistic forecasts can also be evaluated using a distributions-oriented approach. A distributions-oriented approach considers the entire joint distribution of forecasts and observations, $f(p, I)$, where $f$ is the joint probability density function and $I$ is the observed binary inundation outcome (i.e., $I = 1$ if a debris flow actually occurred in the grid cell, else $I = 0$). The joint distribution contains all the relevant information about the forecasts and observations needed for a complete verification of a forecast model (Wilks, 2019). It can be factored in two ways into conditional and marginal probabilities that are more practical for analysis, one of which is the calibration-refinement factorization (Murphy and Winkler, 1987):

$$f(p, I) = f(I \mid p)f(p), \tag{1}$$

This factorization allows inspection of two desirable properties of probabilistic forecasts: (i) they should be reliable in that the forecast event actually happens with a frequency close to the forecast probability, $f(I = 1 \mid p) \cong p$ (e.g., a forecast probability of 30% comes true ~30% of the time); and (ii) the distribution of forecast probabilities $f(p)$ should be dispersed toward the extreme values of zero and one, indicating that the model has confidence in its own predictions (Gneiting et al., 2007; Murphy, 1993). The property of (i) is referred to as calibration (i.e., reliability) and the property of (ii) is referred to as refinement (i.e., sharpness), and a general goal with probabilistic forecast models is for them to be as sharp as possible without sacrificing calibration (Gneiting et al., 2007; Wilks, 2019). We borrowed distributions-oriented methodologies developed in the weather modeling community over the last several decades (e.g., Bröcker and Smith, 2007; DeGroot and Fienberg, 1983; Gneiting et al., 2007; Murphy, 1993; Murphy and Winkler, 1987) to graphically assess the calibration-refinement factorization. Specifically, we used the reliability diagram (Bröcker and Smith, 2007; Wilks, 2019) to separately visualize the calibration and the refinement of the forecast model.

The first component of a reliability diagram is the calibration curve, a function of the conditional distribution $f(I \mid p)$ that provides a visual assessment of the fit between the distribution of forecast probabilities and the observed zones of inundation (Bröcker and Smith, 2007; Gneiting et al., 2007). Quantities on the x-axis answer the question, "what is the mean probability of inundation, $\bar{p}_k$, of all grid cells in the $k^{th}$ bin?" On the y-axis is the frequency of observed inundation conditioned on the binned forecast probabilities, also referred to as the observed relative frequency, which provides an estimate of the calibration distribution: $y_k \approx f(I = 1 \mid p)$ (Murphy and Winkler, 1987). These answer the question, "given a forecast probability of $p$, how often is it correct?" With a perfect probabilistic forecast, the bin-averaged probabilities will exactly match the observed relative frequencies, $y_k = p_k$, and points will fall along the one-to-one line in the calibration curve (Bröcker and Smith, 2007). In reality, sampling variability causes deviations from the one-to-one line even for a perfectly reliable model (Wilks, 2019). Points that fall above the one-to-one line indicate that the model is under-predicting the observed extent of inundation, referred to as under-forecasting, while points below the line indicate over-forecasting. The $(\bar{p}_k, y_k)$ are computed as:

$$\bar{p}_k = \frac{1}{|\omega_k|} \Sigma_{\omega_k \subset \Omega} \, p \,, \tag{2}$$

$$y_k = \frac{1}{|\omega_k|} \Sigma_{\omega_k \subset \Omega} \, I, \tag{3}$$

where $\Omega$ is the set of all grid cells in the spatial modeling domain, $\omega_k$ is the subset of $\Omega$ that satisfies $p \in B_k$, $B_k$ is the $k^{\text{th}}$ bin interval, and $|\omega_k|$ is the number of model grid cells in $\omega_k$. In all cases, we used bin widths of 10 percentage points. Figure S2 provides a visual demonstration of $\omega_k$ and the computation of $y_k$. Forecast reliability was assessed with the mean residual, a measure of bias, and the residual sum of squares, a measure of accuracy, between the $(\bar{p}_k, y_k)$.

The second component of a reliability diagram shows the refinement distribution of the forecasts, answering the question, "how often is each probability of inundation predicted by the forecast model?" This plot shows the histogram of $f(p)$ using the same bins as used in constructing the calibration curve. A sharp forecast will predict probabilities close to zero or one most of the time and will therefore have the highest counts near the boundaries of the histogram. We used the standard deviation of forecast probabilities, $\sigma_p$, as a measure of forecast sharpness because a larger standard deviation indicates greater dispersion toward the extreme values (Bradley et al., 2019).

### 3.7 Sensitivity analyses

We performed two analyses to explore the sensitivity of inundated area to each of the three ProDF input parameters (debris-flow volume, $\chi$, and $\tau_y$). The goals of these sensitivity analyses were to apportion uncertainty in the model output amongst the three parameters and to rank them in terms of relative importance for determining inundated area (Razavi and Gupta, 2015; Saltelli et al., 2008). We used inundated area as a summary of model output because it serves as a simple proxy for downstream impacts. We used the SALib Python package (Herman and Usher, 2017; Iwanaga et al., 2022) to implement the PAWN global sensitivity analysis method (Pianosi and Wagener, 2018). The method returns a sensitivity index between zero and one for each input parameter, with higher indices indicating greater influence on model output.

First, we performed a domain-aggregated sensitivity analysis. To distinguish a significant sensitivity value from one that is due solely to approximation error of the PAWN method, we included a dummy parameter of random numbers in the analysis. Bootstrapping (n=50) was used to compute 95% confidence intervals about the median sensitivity index of each parameter. Parameters whose 95% confidence interval exceeded that of the dummy parameter were considered significant (Pianosi and Wagener, 2018).

Second, we computed spatially distributed sensitivity indices in every grid cell where simulated inundation occurred on a cell-by-cell basis. The sensitivity response variable was taken to be the local binary inundation value, $I$, from each ProDF simulation. This analysis allowed us to compare the relative importance of debris-flow volume, $\chi$, and $\tau_y$ as a function of location, revealing patterns of model sensitivity over the length of the debris-flow runout paths.

## 4 Results

The forecast probability of debris-flow inundation is shown in Fig. 3. Areas that had a non-zero probability of inundation overlapped with 99% of the areas with observed debris-flow inundation, and the region of probabilities exceeding 1% overlapped with 94% of the observed area inundated (Fig. 3a), indicating that the observation was contained within the range of outcomes

represented by the ensemble. However, estimates of area inundated and visual examination of the inundation maps created with different probability thresholds (e.g., 84%, 50%, 16%) indicate that inundation extent was under-forecast (Fig. 4). The highest forecast probabilities are restricted to the main channels of the Montecito, San Ysidro, and Buena Vista Creek Basins. This can be seen by examining the binary inundation map created using a threshold probability of 84% (i.e., $p \geq 0.84$), which resulted in a total inundated area of just 63,000 $m^2$ and a poor similarity index of -0.95 (Fig. 4a). The inundation map created using a threshold probability of 50% also substantially underestimated the extent of the observed inundation, resulting in an inundated area of 760,000 $m^2$ and a similarity index of -0.51 (Fig. 4b). Using a threshold probability of 16% resulted in a binary inundation map that compared best with observations, inundating 3,070,000 $m^2$ and producing a similarity index of -0.02 (Fig. 4c). For comparison, validation of the flow mobility parameter posterior distribution on Montecito Creek resulted in a similarity index of -0.047 (Fig. S3).

The calibration component of the reliability diagram shows that the forecast average probabilities were small relative to the observed relative frequencies, which similarly indicates that the forecast tended to underestimate the observed extent of inundation (Fig. 3b). For example, areas that had a forecast inundation probability of approximately 25%, 50%, and 75% were observed to be inundated at rates of roughly 40%, 75%, and 90%, respectively (Fig. 3b). The mean residual between the binned forecast probabilities and the observed frequencies was -15.9%, with a residual sum of squares of 0.31 (Table 1). The histogram of forecast probabilities indicates a lack of forecast sharpness because the probabilities are clustered near zero and monotonically decrease in frequency toward one (Fig. 3c). Approximately 84% of forecast probabilities were between 0-10% and only 0.1% were between 90%-100%. While the debris-flow volumes (summed across all six basins) ranged from 0 $m^3$ to over 5,500,000 $m^3$ in the ensemble forecast, the median value was 198,000 $m^3$, only 29% of the debris-flow volume observed in the 2018 event.

In contrast, when debris-flow volume was set to the observed value for each basin (Scenario A), analysis of the probabilistic inundation map shows forecast probabilities that were well-calibrated to the observed frequencies (Fig. 5a-b). The mean residual was 0.1% and the residual sum of squares was 0.05, demonstrating a lack of bias and improved reliability (Table 1). The forecast generated from Scenario A was sharp in that the greatest frequency of forecast probabilities were those near zero and one. This was evident visually (Fig. 5b) and statistically, as the standard deviation of inundation probabilities in Scenario A (0.22) was greater than that of the WRF ensemble forecast (0.14), indicating more dispersal toward the extreme values (Table 1). The improvement in sharpness was largely due to a 25-fold increase in the area predicted to have a probability of inundation between 90%-100% relative to the ensemble forecast, as approximately 2.5% of Scenario A inundation probabilities were between 90%-100%.

Scenario B, where we used predictions of debris-flow volume based on observed peak $I_{15}$, resulted in an over-forecast of inundation extent (Fig. 5c-d). Forecast probabilities were generally too high relative to the observed frequencies of inundation as demonstrated by a mean residual of 10.7% in the reliability diagram (Fig. 5d; Table 1). The debris-flow volume summed across all six basins totaled 1,114,000 $m^3$, or 164% of the observed debris-flow volume.

The global sensitivity analysis revealed that the runout model was significantly sensitive to all three parameters (debris-flow volume, $\chi$, $\tau_y$) with volume being the most influential (Table 2). The median PAWN sensitivity indices associated with volume, $\chi$, $\tau_y$, and the dummy variable were 0.38, 0.10, 0.09, and 0.05, respectively. Additionally, maps depicting spatially distributed sensitivity indices indicate that the relative importance of the input parameters varied in both the downstream and across-stream directions (Fig. 6). The $\chi$ parameter had greatest influence at higher elevations in overbank areas removed from the primary

channel, with latitudinally-binned averages that are nearly twice as high near the fan apex when compared to those near the ocean (Fig. 6a). The $\tau_y$ parameter and debris-flow volume showed the opposite pattern, with greater importance at lower latitudes near the distal portion of the fan close to the ocean (Fig. 6b-c), although the flow volume also exerted a strong control on inundation throughout the model domain. Any of the three parameters may be most influential in determining whether an area will be inundated depending on location (Fig. 6d).

**5 Discussion**

Our study results indicate that reducing uncertainty in debris-flow volume predictions will have a substantial effect on reducing uncertainty associated with inundation. Even in the region for which it was developed, the prediction uncertainty associated with the EAV model can exceed an order of magnitude (Gartner et al., 2014; Barnhart et al., 2021), with greater uncertainties likely when models are applied in landscapes that differ from where they were trained (e.g., Gorr et al., 2023; Rengers et al., 2023). In the absence of sufficient data to train local or regional postfire debris-flow volume models, it may be possible to apply a correction factor to an existing model that was trained elsewhere (e.g., Rengers et al., 2023). In addition, variations in forecast storm structure can result in highly variable precipitation intensity, duration, timing, and spatial distribution (Fig. 1c) (Oakley et al., 2023), driving further uncertainty in debris-flow volume prediction through the forecast peak $I_{15}$. This uncertainty propagates from rainfall into a wide range of inundation outcomes (Fig. S4). Improved prediction of postfire debris-flow volume, whether through improved forecasts of $I_{15}$ or improved volume models, would reduce the uncertainty associated with the inundation probabilities and result in a narrower range of inundation outcomes. However, the modular structure of the framework presented here makes it straightforward for the debris-flow likelihood and volume models to be replaced with updated or region-specific alternatives as they become available (e.g., Diakakis et al., 2023; Nyman et al., 2015; Santi and Morandi, 2013; Staley et al., 2013; Wall et al., 2023).

Calibration of the flow mobility parameters created a posterior distribution with a range of support over $\chi$-$\tau_y$ space (Fig. S3a). Capturing the effect of this spectrum of possible flow behaviors was important because the debris-flow properties that we expect to influence the flow mobility parameters (e.g., grain size distribution, sediment concentration) are unknown before an event. Furthermore, it is common for debris-flow properties to change as flows move downslope (Iverson, 1997, 2003). ProDF uses constant values for $\chi$ and $\tau_y$ across the simulation domain, limiting debris-flow behavior to a single characteristic type. The forecast model, however, enabled the representation of multiple flow rheologies in the probabilistic prediction of inundation by sampling flow mobility parameters from the calibrated posterior distribution. At sites without past events to aid in calibration, we would expect greater uncertainty in any estimate of a posterior parameter distribution. Gorr et al. (2023) found that the best fit calibrated value of yield strength for a small debris flow in northern Arizona was more than three times greater relative to that calibrated for the Montecito debris flows in southern California. Poor constraints on yield strength in forward modeling applications could result in greater uncertainty in predictions of inundation, particularly in terms of total downstream travel distance since the yield strength plays an important role in determining when the modeled flow comes to rest (Fig. 6b). Still, the greater importance of debris-flow volume in determining inundated area indicates that placing better constraints on debris-flow volume, including variations resulting from entrainment or deposition along the travel path, warrants prioritization over constraints on flow mobility parameters (Fig. 6 and Fig. S4; Table 2).

Analysis of the forecast inundation probabilities using the reliability curve showed that the model tended to under-forecast the observed frequencies of inundation (Fig. 3b). For example, in areas with a forecast inundation probability of 40-50%, the frequency

of observed inundation was approximately 70%. In other words, the model simulated inundation less often than observed. We attribute this bias, at least in part, to the extreme nature of the 2018 rainfall event and the challenges of representing this event at a 24-hour lead time in the atmospheric model. We expect that the large spread in peak $I_{15}$ in the WRF atmospheric model forecast and the uncertainties associated with it are not unique to our study area or the modeled rainstorm and should be considered in future applications of this and similar probabilistic debris-flow inundation model frameworks. Uncertainties in mesoscale precipitation forecasts of short duration, high intensity rainfall (e.g., peak $I_{15}$) are well documented even at lead times shorter than 24 hours (Cannon et al., 2020; English et al., 2021). The observed peak $I_{15}$ values lie in the tail of the atmospheric model ensemble spread (Fig. 1c) (Kean et al., 2019; Oakley et al., 2023); thus, most $I_{15}$ values in the ensemble are lower than observed. This leads to lower ensemble predictions of debris-flow volume and likelihood with the EAV and M1 models and therefore less inundation than would be expected given the observed $I_{15}$. This may also explain why the best match of simulated inundation to the observed deposits occurs at a threshold probability level of 16% while threshold probabilities of 50% and 84% resulted in substantial underestimates of area inundated (Fig. 3 and Fig. 4).

Our interpretation that the ensemble distribution of predicted $I_{15}$ led to under-forecasting is supported by comparison with the reliability diagrams associated with forecast scenarios A and B, which were run using observed debris-flow volumes and observed peak $I_{15}$, respectively (Fig. 5). The calibration curves from these two scenarios indicate high sensitivity of the calibration to the input debris-flow volumes, which are influenced by peak $I_{15}$. When the volumes predicted from $I_{15}$ were too low, as in the fully predictive model, the calibration curve lies above the one-to-one line, indicating under-forecasting (Fig. 3b). The calibration curve passes through the one-to-one line when the observed volumes, which are greater than those computed using the ensemble predictions of $I_{15}$, were used (Scenario A; Fig. 5b). Finally, the calibration curve drops below the one-to-one line, indicating over-forecasting, when volumes are computed based on the observed $I_{15}$ (Scenario B; Fig. 5d and Fig. S1). Volumes computed from the observed $I_{15}$ were greater than both the observed volumes and those computed from the ensemble predictions of $I_{15}$. As a result, the model would have over-predicted inundation area if the atmospheric model yielded a perfect prediction of peak $I_{15}$.

While the sum of predicted volumes from Scenario B was 164% of the total observed volume, this amount of error is within the range of what is expected. The EAV model predicts the natural logarithm of volume with a standard error of 1.04 (Gartner et al., 2014), which translates to a 95% probability that the observed volume will be between 13% and 770% of the modeled value (Barnhart et al., 2023). This degree of uncertainty highlights the potential gains of improving models for postfire debris-flow volume. When the observed volumes were used (i.e., scenario A), the calibration curve is close to the diagonal, and the refinement distribution shows that extreme values were most commonly forecast (Fig. 5b). This indicates that the calibrated forecast model is both reliable and sharp when the volumes are well-constrained. However, in the forecast model where debris-flow volume was a function of the peak $I_{15}$ derived from the atmospheric model ensemble, the ultimate effects of having a lower-than-observed peak $I_{15}$ in many atmospheric model ensemble members was likely at least partially offset by the EAV model's bias to overpredict debris-flow volumes (Barnhart et al., 2021; Kean et al., 2019).

The methods presented here take a step toward near-real time assessments of postfire debris-flow hazards associated with an incoming rainstorm. Our work builds on that of Oakley et al. (2023), who used the same atmospheric model ensemble to produce probabilistic predictions of debris-flow likelihood and volume in watersheds burned by the 2017 Thomas Fire. They did not include predictions of postfire debris-flow inundation, but they identified that a product linking together postfire debris-flow volume ensembles with runout models was an important area of focus for future research to support impact-based decision making (Oakley et al., 2023). Further, recent surveys demonstrate a need for hazard assessment products that connect debris-flow inundation models

with forecasts of rainfall in the short period of time between fire containment and the first precipitation event (Barnhart et al., 2023; Gourley et al., 2020). Considering that decision quality improves when probabilistic information is presented appropriately in weather forecasts (Ripberger et al., 2022), the types of maps generated by the model framework presented here could be used to support decisions regarding evacuations, staging of equipment and emergency personnel, and debris-flow mitigation efforts.

Additional assessments of the integrated modeling approach presented here in different geographic and climatic settings would help generalize findings and develop guidelines for constraining flow mobility parameters in areas where there are no historical observations that can be used to calibrate the runout model, ProDF. Computing resource constraints present a challenge for future studies and real world use because the probabilistic forecast of debris-flow inundation and the atmospheric ensemble forecast both require many core-hours of computing time. Approaches to reduce computation times include optimizing aspects of the simulation for the task at hand (e.g., number of ensemble members, horizontal grid spacing; Oakley et al., 2023), running debris-flow runout simulations massively in parallel, and limiting the spatial extent of modeling efforts. The framework proposed here could also be applied in a pre-fire context to assess postfire hazards. In this case, the rainfall intensity input could be determined from local climatological data and soil burn severity characteristics could be simulated (e.g., Kean and Staley, 2021; Staley et al., 2018; Wells et al., 2023). Pre-fire assessments of postfire hazards provide valuable insight into areas of greatest concern that could assist with community planning, emergency management, and debris-flow hazard mitigation (McCoy et al., 2016; Tillery et al., 2014).

## 6 Conclusions

We created a computational framework for probabilistic predictions of rainfall induced debris-flow inundation downstream of burned basins that integrates an ensemble forecast of rainfall with existing models for postfire debris-flow likelihood, volume, and runout. We applied this methodology by using a 24-hour, 100-member atmospheric model ensemble forecast of rainfall intensity associated with a destructive debris-flow event that followed the 2017 Thomas Fire. When debris-flow volumes were well-constrained, the probabilistic model predictions were sharp and well-calibrated to the observed area inundated. In the fully predictive model, approximately 99% of the observed inundation area was contained within a region where the simulated probability of inundation was greater than zero. In general, however, we found that the model under-forecasted the area inundated. We attribute the under-forecasting of inundation extent to the fact that the observed peak 15-minute rainfall rates were in the upper tail of the atmospheric model ensemble distribution of forecast rainfall rates.

A sensitivity analysis indicated that debris-flow volume had the greatest influence on the simulated area inundated, while the two flow mobility parameters had a lesser but still significant influence. A spatially distributed sensitivity analysis showed that the importance of flow volume and the flow-mobility parameters varied across the model domain in systematic ways, and it showed that each parameter may be the most important locally depending on location within the landscape. Future efforts to constrain flow mobility parameters in a range of postfire settings would assist with reducing uncertainty in debris-flow runout model predictions, but the greatest gains in model performance are likely to result from improving estimates of debris-flow volume. Application of the proposed framework to other sites with different topographical and climatological properties would help assess the generalizability of findings related to parameter sensitivity. While we focus on an application using forecast rainfall, the proposed framework could also be used to assess postfire debris-flow hazards in response to design rainstorms before a fire starts.

*Data availability*. The model code, input data, and output data used in this study are available in a Zenodo online archive under the open-access Creative Commons Attribution 4.0 International license (Prescott et al., 2023).

*Author contributions*. LAM, K-SJ, and NAO acquired funding for this project. LAM, K-SJ, and ABP conceptualized and developed the methodology. ABP developed the model code and ran simulations, with contributions from KRB. ABP and LAM prepared the manuscript draft, and all co-authors reviewed and edited the draft.

*Competing interests*. The contact author has declared that none of the authors has any competing interests.


*Disclaimer*. Any use of trade, firm, or product names is for descriptive purposes only and does not imply endorsement by the U.S. Government.

*Acknowledgments*. This work was supported by the Joint Fire Sciences Program through grant #L20AC00029 and the California
Department of Water Resources Atmospheric River Program (4600013361). This material is based upon High Performance Computing (HPC) resources supported by the University of Arizona TRIF, UITS, and Research, Innovation, and Impact (RII) and maintained by the University of Arizona Research Technologies department. The authors are grateful to Matthew Simpson at Scripps Institution of Oceanography for running the WRF simulations used in this research and Brady Gales at the University of Arizona for early contributions to the experimental design.

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

**Figures**

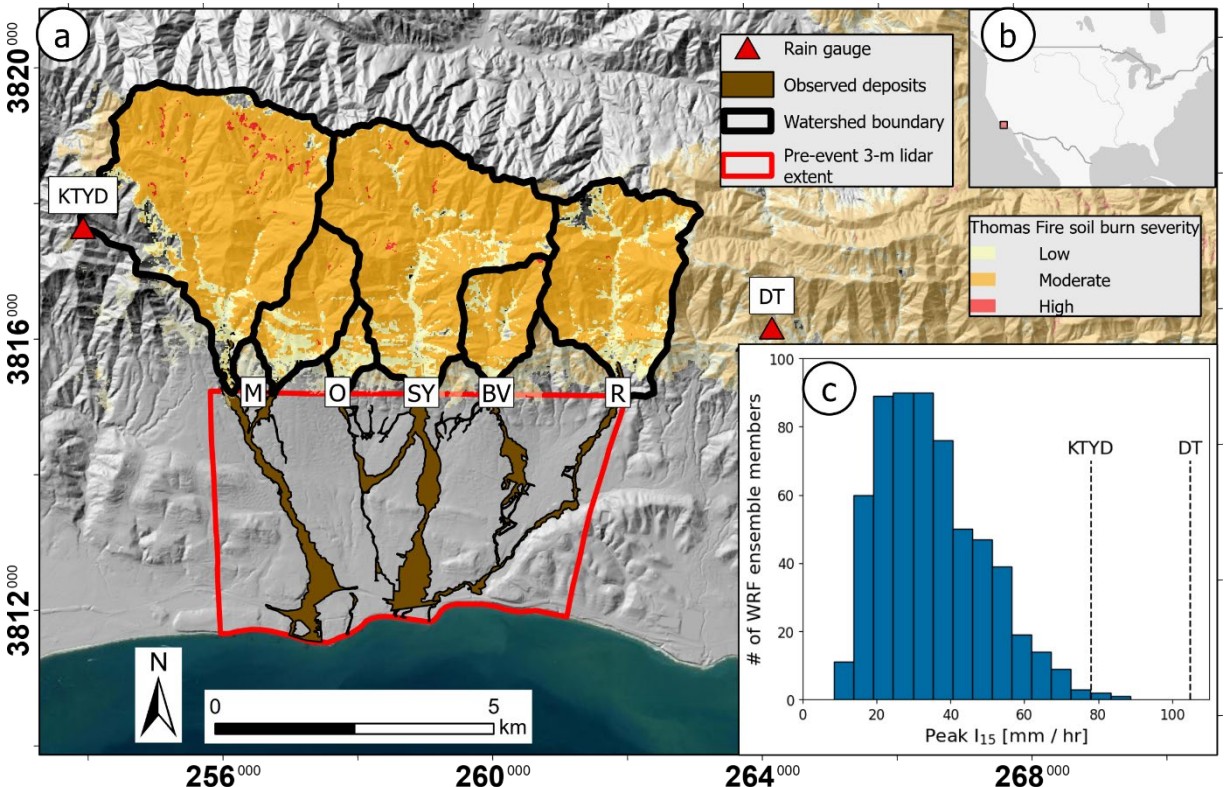

Figure 1: (a) Site map of the 9 January 2018 Montecito debris-flow event. Burned basins drain into (M) Montecito, (O) Oak, (SY) San Ysidro, (BV) Buena Vista, and (R) Romero Creeks (note that 2 burned basins drain into Montecito Creek); (b) Site location in context of the USA; (c) histogram of peak $I_{15}$ extracted from the atmospheric model ensemble. Dashed lines show observations at the KTYD and Doulton Tunnel (DT) rain gauges. Ticks along the boundaries of (a) give coordinates in NAD 1983 UTM zone 11N. In this figure and in all following maps, the base map was sourced from ESRI and the U.S. Department of Agriculture Farm Services Agency, the hillshade layer was generated from the 10-m resolution National Elevation Dataset (U.S. Geological Survey, 2020), and the soil burn severity layer was sourced from U.S. Forest Service (2020).

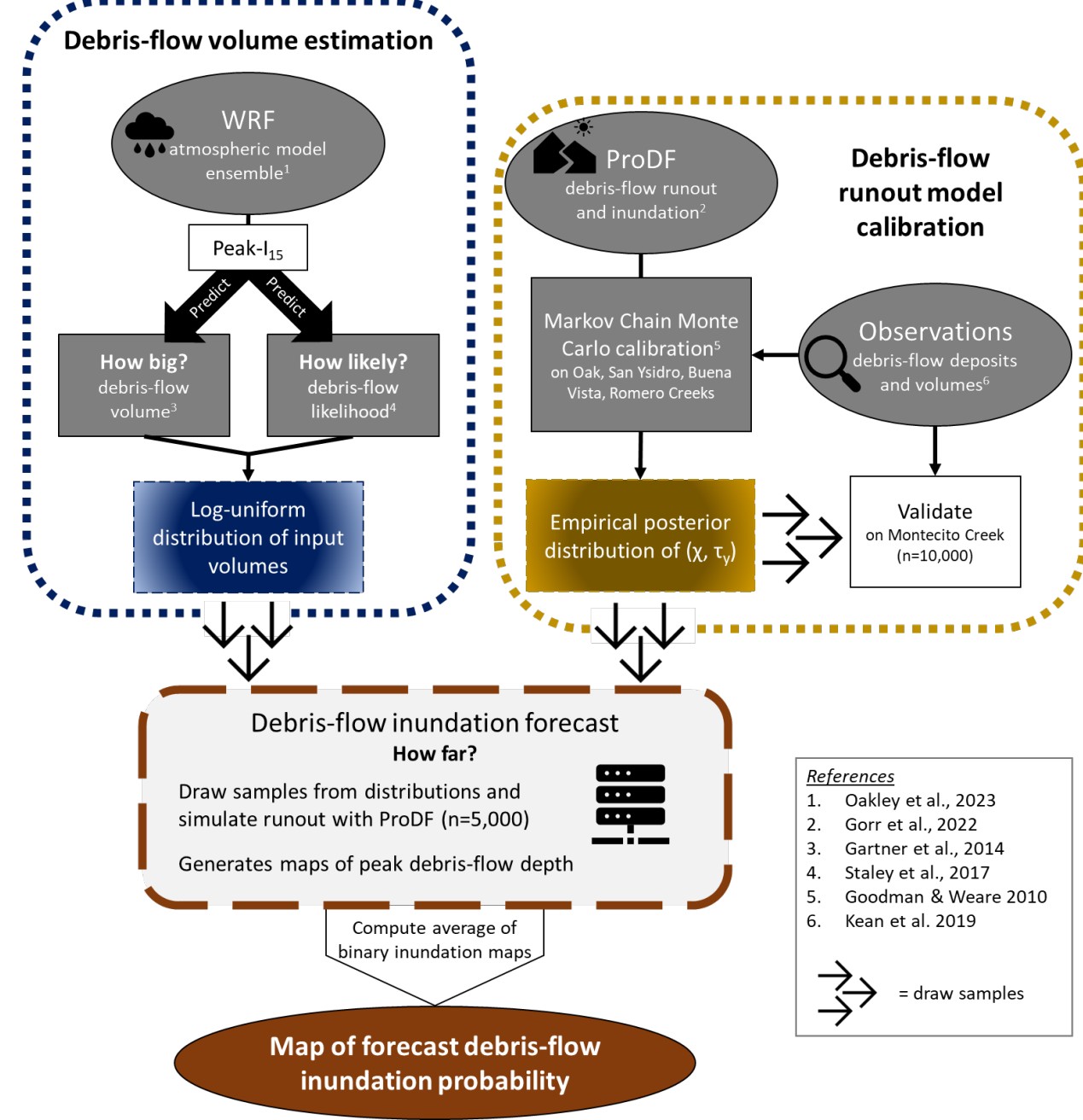

**Figure 2: Schematic of the probabilistic debris-flow inundation forecast model.**

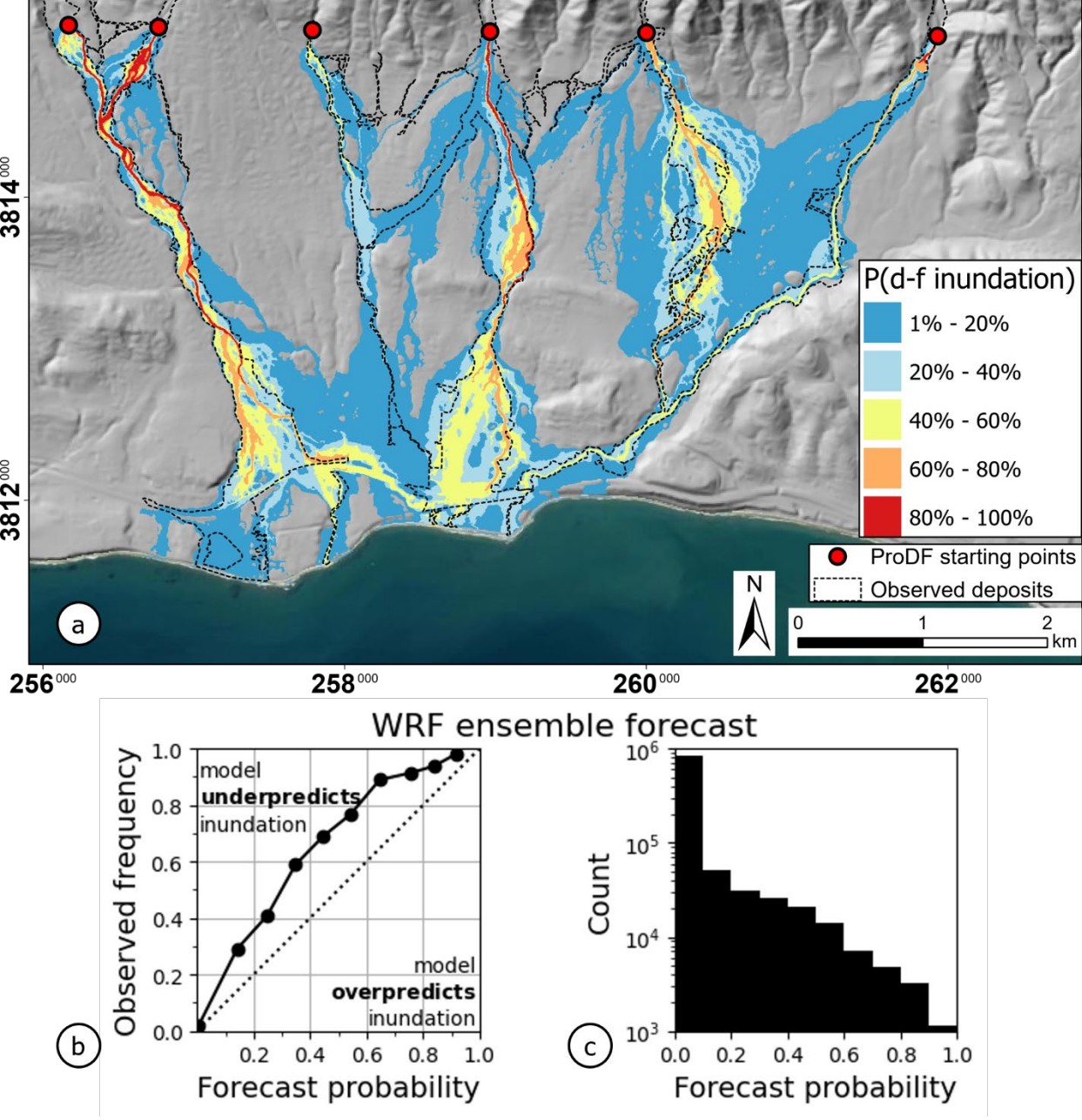

Figure 3: Results from the debris-flow inundation simulation driven with the 100-member WRF ensemble predictions of peak $I_{15}$. (a) This map shows the forecast probabilities of debris-flow inundation, P(d-f inundation), in all model cells with probability greater than or equal to 1%. Dashed lines show the extent of observed debris-flow deposits. Ticks along the map boundaries give coordinates in NAD 1983 UTM zone 11N. (b & c) The two-part reliability diagram shows the calibration-refinement factorization of the joint distribution of forecasts and observations (described in Section 3.6). In the calibration curve (b), a perfectly calibrated model will lie along the 1:1 line, and points above (below) the diagonal indicate that the model is under-forecasting (over-forecasting) the observed frequency of inundation. The histogram in (c) demonstrates the refinement distribution of the forecast probabilities. A sharp forecast will have the highest count of probabilities toward the extreme values of 0 and 1.

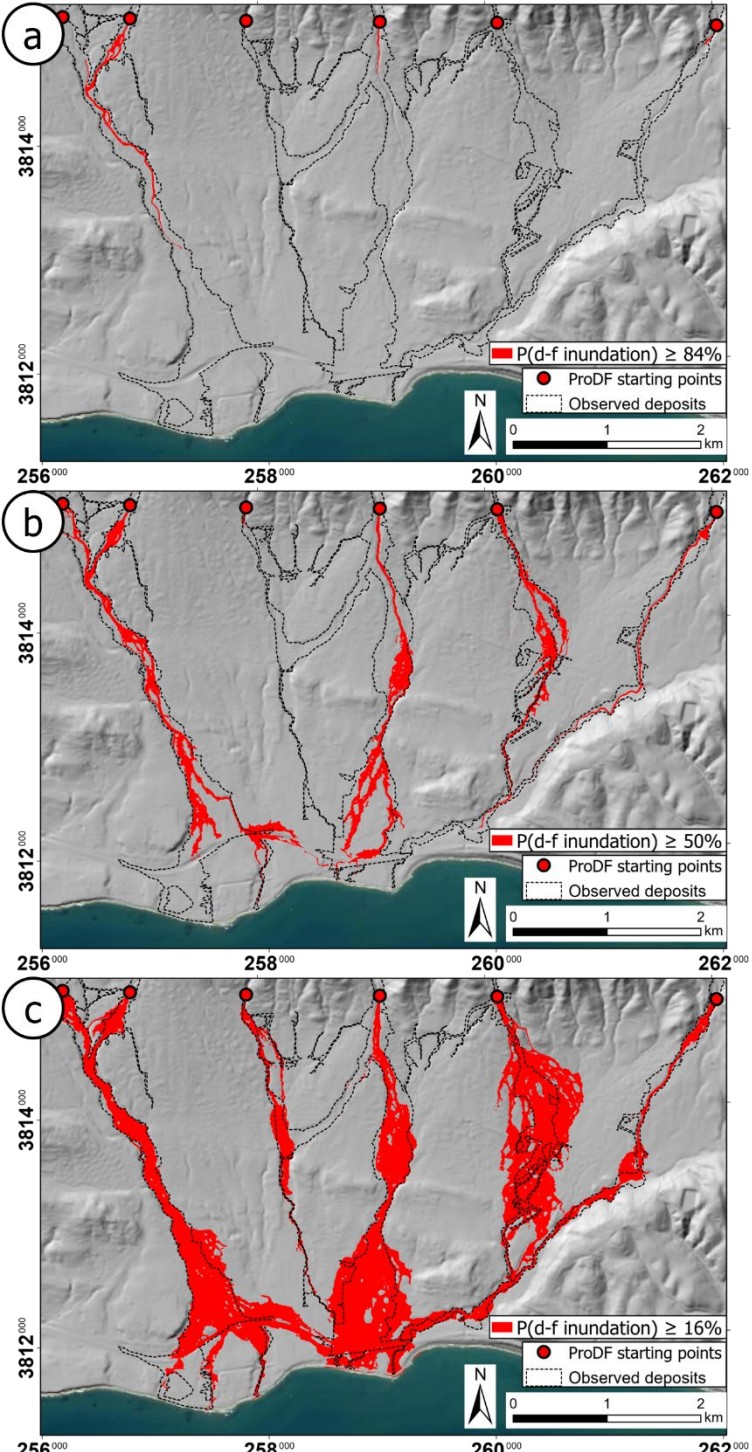

Figure 4: Binary maps of area inundated created by thresholding the forecast probabilities of debris-flow inundation, P(d-f inundation), at (a) 84%, (b) 50%, and (c) 16%. A grid cell is marked as inundated when the local forecast probability exceeds the threshold value. Inundated area increases as the probability threshold decreases, producing similarity indices of -0.95, -0.51, and -0.02, respectively. Ticks along the boundaries of each map give coordinates in 20 NAD 1983 UTM zone 11N.

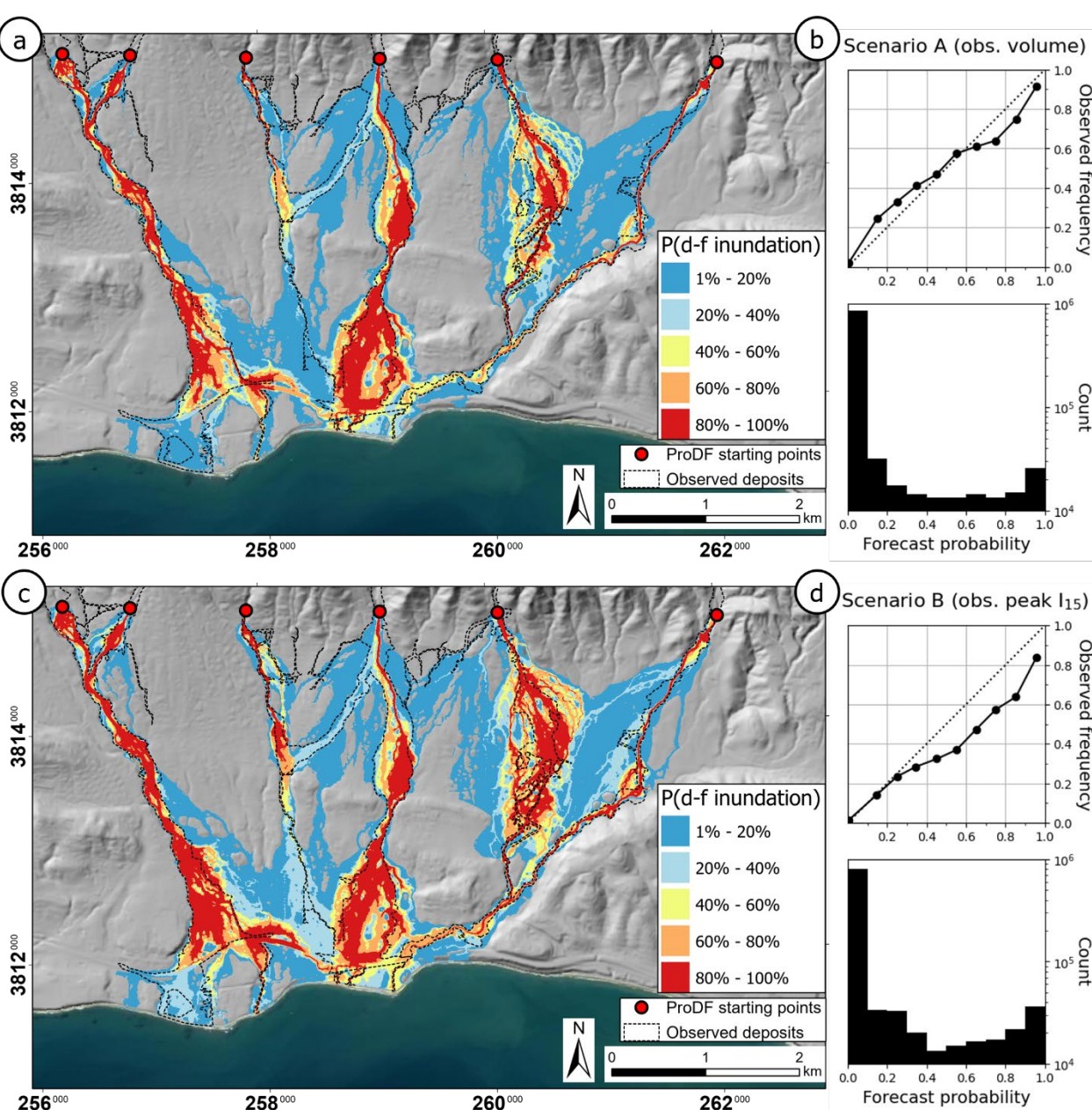

Figure 5: Inundation probability maps and two-part reliability diagrams resulting from the constant-volume simulation scenarios A and B; see Fig. 3 caption for panel descriptions. (a-b) Scenario A used the measured input debris-flow volumes for each drainage basin (Kean et al., 2019). (c-d) Scenario B used input debris-flow volumes predicted with the EAV model using $I_{15}$ values interpolated to each ProDF starting point from observations at the KTYD and Doulton Tunnel rain gauges (78 and 105 mm h$^{-1}$, respectively; Kean et al., 2019). Ticks along the boundaries of each map give coordinates in NAD 1983 UTM zone 11N.

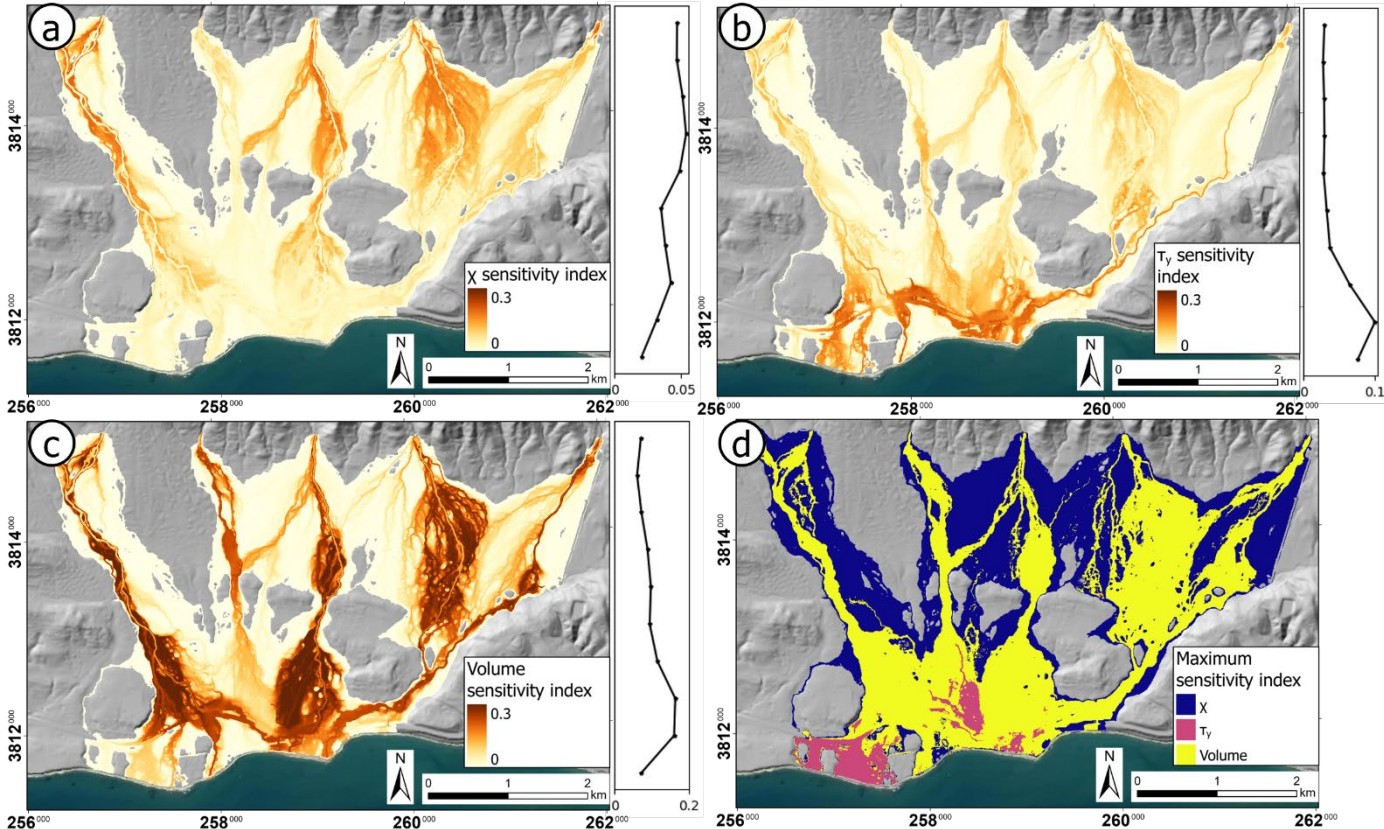

**Figure 6: Spatially distributed PAWN sensitivity indices for forecast inundation against the model parameters (a) χ; (b) $\tau_y$; and (c) debris-flow volume; and (d) the parameter that resulted in the maximum sensitivity index in each grid cell. The vertical plots on the right of panels (a) through (c) show the longitudinally-averaged sensitivity index as a function of binned latitude, demonstrating how parameter importance varies with distance from the upstream basin outlets. Depending on location in the simulation domain, each of the input model parameters may be the most influential. Ticks along the boundaries of each map give coordinates in NAD 1983 UTM zone 11N.**

**Tables**

| | WRF ensemble forecast | Scenario A | Scenario B |
|---|---|---|---|
| Bias, $\overline{\overline{p}_k - y_k}$ | -15.9% | 0.1% | 10.7% |
| Accuracy, $\sum(\overline{p}_k - y_k)^2$ | 0.31 | 0.05 | 0.18 |
| Sharpness, $\sigma_p$ | 0.14 | 0.22 | 0.25 |

665

**Table 1: Summary metrics of the reliability diagram for each debris-flow inundation simulation. See Section 3.6 for definitions.**

| Parameter | Median PAWN sensitivity index (2.5$^{th}$ – 97.5$^{th}$ percentile values) |
|---|---|
| $\chi$ | 0.098 (0.082 – 0.111) |
| $\tau_y$ | 0.094 (0.080 – 0.113) |
| debris-flow volume | 0.382 (0.360 – 0.403) |
| dummy | 0.049 (0.040 – 0.059) |

**Table 2: PAWN sensitivity indices (Pianosi and Wagener, 2018) with 95% confidence intervals from bootstrapping (n=50 iterations).**