# Peer review of "Probabilistic assessment of postfire debris-flow inundation in response to forecast rainfall"

_EGUsphere, 2023_

## Author Response (AR1)

In this response document, all line numbers refer to the updated manuscript (without track-changes) unless stated otherwise.

**Comments by Reviewer 1, anonymous** https://doi.org/10.5194/egusphere-2023-1931-RC1

I suggest the manuscript be rejected for the following reasons:

- The paper is an incomplete work. As stated by the authors, "This study is a first step toward the development of an operational framework for probabilistic assessments…". Most of the main texts, pages 3-7, are describing the methods, only half of a page is demonstrating the results, and one-half of a page is demonstrating the discussion. No innovation result/conclusion may be found in the manuscript. The paper may be rewritten and then resubmitted after the research work has been completed.

  Response

  We think that the work presented in this manuscript represents a complete study. We welcome this opportunity to clarify our objectives and to clarify some of the language used in the initial submission. Our use of the term "operational" is drawn from the context of the National Oceanic and Atmospheric Administration (NOAA) Readiness Levels (https://orta.research.noaa.gov/support/readiness-levels/). The nine NOAA Readiness Levels provide a systematic way for determining the maturity of a research project, from theoretical development (Readiness Level 1) through the various stages of project development (including experimentation, evaluation, user documentation, etc.) up to full deployment and routine use of the project by the intended audience (Readiness Level 9). In this context, the term "operational" is reserved for methods or tools in the final stages of development that have been deployed in a near-real world environment (Readiness Levels 7-9). As a proof-of-concept and feasibility study, this paper falls in the early phase of development of a method and would be classified in Readiness Level 3.

  This work is an important and necessary step towards a fully operational (Readiness Level 9) rapid postfire debris-flow inundation hazard assessment tool, but there is far more intermediate work to accomplish before such a tool may be regularly used for decision making. Without this modeling framework linking together the various rainfall and debris-flow prediction components, we have no way of understanding the sensitivies of the model output to each of the input parameters. Attempting to go from Readiness Level 1 to a Readiness Level 9 product in one publication is not feasible, not typical, and would be unable to resolve all the open research questions that stand in the way of such a product. Our work is therefore an initial and critical step towards a product applicable in the real world.

  To clarify the objectives of our study, we have chosen to remove the term "operational" from the manuscript entirely and replace it with "near-real time" (lines 25, 92, 351), "real world" (line 358), and "rapid hazard assessment framework" (line 65). In the Abstract, we have replaced the sentence quoted by the reviewer with (lines 24-27): "This study represents a first step toward a near-real time hazard assessment product that includes probabilistic estimates of debris-flow inundation and provides guidance for future improvements to this and similar model frameworks by identifying key sources of uncertainty."

In order to better highlight the results and conclusions of this work, we have rewritten the Conclusions section entirely. We describe this further in the response to the reviewer's second comment.

- The structures of the abstract and conclusions are also strange. For example, in the part Conclusions, it is concluded in the first sentence: "Probabilistic debris-flow inundation maps may support emergency management and hazard mitigation efforts in advance of forecasted storms over burned areas". Have you given enough evidences to verify this conclusion? The second sentence of the part, "To explore the feasibility of producing such maps, we integrated output…, is describing the methods, which is not encouraged to emerge in the part Conclusions.

Response

We have rewritten the Conclusions section to better highlight the results and main takeaways from our study. This includes removal of the lines that the reviewer has quoted. A summary of the primary contributions and conclusions of this study are:

1. We created a computational framework for probabilistic predictions of rainfall induced debris-flow inundation downstream of burned basins that integrates an ensemble forecast of rainfall with existing models for postfire debris-flow likelihood, volume, and runout.
2. We applied this framework using a 24-hour, 100-member atmospheric model ensemble forecast of rainfall intensity associated with a destructive debris-flow event that followed the 2017 Thomas Fire. Approximately 99% of the observed inundation area was contained within a region where the simulated probability of inundation was greater than zero.
3. Debris-flow volume had the greatest influence on the simulated area inundated, while the flow mobility parameters had a lesser but still significant influence.
4. Reducing uncertainty in predictions of postfire debris-flow volume and predictions of rainfall intensities over sub-hourly durations (e.g. 15 minutes), a key input for postfire debris-flow volume models, would lead to the greatest gains in model performance.
5. Future work towards a near-real time product aimed at enhancing decision support for postfire debris-flow hazards may build on this work by exploring the role of alternative modeling framework components and the performance of the framework when applied to diverse locations.

In addition, we have added the following sentences to the Abstract (lines 16-22) to highlight our results and main contributions:

"Approximately 94% of the observed inundated area was forecast to have an inundation probability greater than 1%, demonstrating that the observed extent of inundation was generally captured within the range of outcomes predicted by the model. In general, however, the model under-forecasts the inundation area, which we attribute primarily to under-forecasting of the peak fifteen-minute rainfall intensity, $I_{15}$, by the atmospheric model ensemble. Peak $I_{15}$ is a key control on debris-flow likelihood and volume. Under-forecasting of $I_{15}$ therefore led to lower than observed flow volumes, which translated into under-forecasting of the inundation area. In contrast, numerical experiments of debris-flow inundation using observed debris-flow volumes demonstrated that the model is reliable and sharp."

With regard to the need for the modeling framework presented in this study, we point to recent publications that surveyed professionals involved in postfire flooding and debris-flow emergency

management. A workshop convened in 2019 brought together over 40 participants from governmental, non-governmental, emergency management, and academic institutions for the purposes of assessing the state of the science in postfire hydrology and to guide future efforts to improve society's ability to mitigate postfire hazards (Gourley et al., 2020). Participants in this workshop identified the long-term goal of developing methodologies that link spatial estimates of rainfall intensity with surface hydrological models to rapidly map potential debris-flow runout paths in real time. This goal, originally stated by a 2005 NOAA and U. S. Geological Survey (USGS) joint task force, remains unmet today (Gourley et al., 2020; NOAA-USGS Debris Flow Task Force, 2005).

In addition, a recent user needs assessment surveyed postfire emergency management professionals in Southern California on the topic of postfire debris-flow inundation hazard products (Barnhart et al., 2023). Participants expressed a perceived need for a real time product that can map potential postfire debris-flow inundation hazards in response to forecast rainfall, and they expressed that such a product should convey a measure of the forecast uncertainties. We discuss these points in the text of the original submission on lines 43-45 and line 65. Finally, we note that probabilistic forecasts generated from ensembles can improve decision making quality when the probabilistic information is presented appropriately (Ripberger et al., 2022). We have modified the manuscript to include the conclusions of Gourley et al. (2020) on lines 52-54:

> "Rapid postfire hazard assessments (e.g., Staley et al., 2016) do not currently provide information about downstream impacts, although recent debris-flow events (Kean et al., 2019) and surveys of the postfire hazards emergency management community highlight the need for such a product (Barnhart et al., 2023; Gourley et al., 2020),"

and Ripberger et al. (2022) on lines 352-354:

> "Considering that decision quality improves when probabilistic information is presented appropriately in weather forecasts (Ripberger et al., 2022), the types of maps generated by the model framework presented here could be used to support decisions regarding evacuations, staging of equipment and emergency personnel, and debris-flow mitigation efforts."

Motivated by this comment as well as comments from Reviewer 2, we moved several figures from the supplement to the main text. One of these figures (Fig. S5 in the original submission, now Table 2) displays the results of the global PAWN sensitivity analysis, which makes it easier to describe how flow volume and flow mobility parameters influence modeled area inundated. Results of the global and spatially distributed sensitivity analyses provide guidance for future work by isolating components of the framework (i.e., models for debris-flow volume) where improvements are likely to lead to the greatest gains in overall model performance. We include additional discussion in the revised manuscript that focuses on the implications of our sensitivity analysis and provides guidance for improving this and similar computational frameworks for postfire debris-flow hazards. The added content on lines 313-317 is:

> "Gorr et al. (2023) found that the best fit calibrated value of yield strength for a small debris flow in northern Arizona was more than three times greater relative to that calibrated for the Montecito debris flows in southern California. Poor constraints on yield strength in forward modeling applications could result in greater uncertainty in predictions of inundation, particularly in terms of total downstream travel distance since the yield strength plays an important role in determining when the modeled flow comes to rest (Fig. 6b)."

On lines 341-345, it is:

> "While the sum of predicted volumes from Scenario B was 164% of the total observed volume, this amount of error is within the range of what is expected. The EAV model predicts the natural logarithm of volume with a standard error of 1.04 (Gartner et al., 2014), which translates to a 95% probability that the observed volume will be between 13% and 770% of the modeled value (Barnhart et al., 2023). This degree of uncertainty highlights the potential gains of improving models for postfire debris-flow volume."

- The title of the paper shown in the webpage, "Online repository for code and data used in: Probabilistic assessment of postfire debris-flow inundation in response to forecast rainfall", is different to that in the manuscript, "Probabilistic assessment of postfire debris-flow inundation in response to forecast rainfall".

    Response

    As far as we are able to tell, the correct titles appear in the expected places in both our submission materials and on the EGUsphere online portal. The title of the online repository that hosts all of the modeling code and data used in our submission, "Online repository for code and data used in: Probabilistic assessment of postfire debris-flow inundation in response to forecast rainfall," appears correctly under the *Assets for review: Model code and software* section in our display of the EGUsphere research article record. We followed the NHESS submission directions when preparing this repository and when citing it within our submission. The title of the repository is necessarily different from the title of the manuscript, "Probabilistic assessment of postfire debris-flow inundation in response to forecast rainfall," so as to prevent confusion.

**Comments by Reviewer 2, Paul Santi** https://doi.org/10.5194/egusphere-2023-1931-RC2

Well written paper based on a solid modeling database.

Figure 3. Should the red circle be "D-f deposit initiation point" since it is where the deposit starts and not the beginning of the actual debris flow

> Response

> The reviewer is correct, and we have edited Fig. 3 (and all other relevant figures) to refer to these as "ProDF starting points," consistent with Gorr et al. (2022).

Figure 3 - any statistical quantification of the quality of the prediction?

> Response

> We think that the best way to evaluate the quality of the probabilistic predictions is through the reliability diagrams because they represent the full joint distribution of the observations and predictions (Wilks, 2019). To aid the reader in assessing the prediction quality, we have moved the reliability diagrams associated with the WRF ensemble forecast of debris-flow inundation into a figure panel with the map of forecast probabilities (formerly Fig. 4a-b, now Fig. 3b-c). Further motivated by comments from the editor, we have added a table to present statistics that measure the bias, calibration accuracy, and sharpness of each simulation (Table 1). These statistics are further described in the Results section.

> If the reviewer desires a single metric quantifying the prediction quality, then the Brier score may be used (Wilks, 2019). The Brier score is essentially the normalized sum of squares of the difference in every grid cell between the observed inundation (i.e., 0 or 1) and the forecast probability (i.e., a real number between 0 and 1). More accurate forecasts produce lower Brier scores. The Brier score of the WRF ensemble forecast is 0.055, that of Scenario A is 0.047, and that of Scenario B is 0.052. However, we decided not to introduce the Brier score into the revised manuscript since we think that the quality of the prediction is best assessed using the reliability diagram, and we hesitate to introduce an additional model performance statistic into an already crowded Methods section.

Figure 5 - axis labels hard to read at that font size

> Response

> We have increased the font size where needed in all figures to improve readability.

Line 233ff - I think it might help to show the p=16% map. When I look at Figure 3, it appears in places that true runout matches best with p>40-60 (Buena Vista) and other places it matches better with p<20 (Montecito). In my opinion, Figure 3 is not very convincing of the accuracy of the forecast system.

> Response

> We have moved the panel figure of binary inundation maps from the supplement into the main text (formerly Fig. S3, now Fig. 4), and we have added additional text describing this panel figure to the Results section (lines 250-256).

We emphasize that the method we present does not rely on optimizing any of the model components in an attempt to fit the simulations to the observations. For example, we use a 24-hour 100-member ensemble to determine the 15-minute rainfall intensities that are used for determining debris-flow likelihood and volume. Accordingly, we think that the appropriate way to assess model performance using Fig. 3 is to compare the observed inundation with the simulated area inundated. Fig. 3 shows that 94% of the observed area inundated was contained within a region where the probability of inundation exceeded 1% and that value increases to 99% when considering the region of all probabilities greater than zero. We conclude that the observation was contained within the range of inundation scenarios represented by the ensemble forecast. In general, we find that the model under-forecasts inundation extent. We explore the underlying cause of this by comparing forecast results with two alternative scenarios, which we refer to as Scenarios A and B, in which debris-flow volumes are defined based on the observed volume and the observed rainfall intensity (rather than the rainfall intensities from the atmospheric model ensemble). Based on these numerical experiments, we determined that we could attribute the under-forecast of inundation extent primarily to the under-forecasting of peak 15-minute rainfall intensities in the atmospheric model ensemble. The reliability diagram for Scenario A demonstrates that when the debris-flow volume is well known, the forecast is well-calibrated (in that points on the calibration curve lie close to the one-to-one line) and refined (a.k.a. sharp, in that probabilities close to zero and one are predicted most often).

I found it a bit difficult to follow the graphical story at times because there was so much reliance and reference to supplemental figures. I realize that the publisher may limit the number of figures included with the paper, but it would help to look for ways that the reader could still understand the ideas and be convinced of the reliability of the model without needing to refer to supplemental figures.

Response

We have moved multiple figures from the supplement into the main text to address the reviewer's point and improve the graphical story told by the manuscript. Specifically, we moved Fig. S3 (now Fig. 4), Fig. S5 (now Table 2), and Fig. S7 (now Fig. 5) into the main text. We also incorporated the reliability diagram panels of Fig. 4 into the new Fig. 3 and Fig. 5 for improved connection between each probabilistic map and the corresponding reliability diagrams. We also added latitudinally-binned averages of the sensitivity indices to the maps of spatially-distributed sensitivity indices to clarify patterns as a function of distance along the fan (formerly Fig. 5a-c, now Fig. 6a-c).

**Comments by Editor, Paolo Tarolli**

Dear authors,

Your article has been revised by two reviewers who proposed rejection and minor changes. You provided a detailed reply in the public discussion. From my side, I would recommend a more substantial description of the results and quantification of bias in the analysis carried out.

You, therefore, can resubmit your work after a careful check and improvements.

Best regards

Paolo Tarolli

NHESS Executive Editor

Response

We are grateful to the editor and the reviewers for their comments and suggestions that have improved the quality of the manuscript considerably. In order to clarify the main results of our study and to further support our discussion of forecast bias and uncertainty, we have expanded the Results section, and we have introduced additional statistical metrics to quantify the bias, accuracy, and sharpness of each of the forecast simulations. In summary, the additions are:

1. Additional descriptions of the probabilistic debris-flow inundation forecast generated using the WRF ensemble of atmospheric models, including: the percent of observations contained within the region of non-zero (and >=1%) inundation probabilities; the varied inundation outcomes when thresholding forecast probabilities by three values (i.e., 84%, 50%, 16%) and the similarity indices and total simulated inundated area associated with each (lines 246-256).
2. An expanded description of the calibration curve including statistical analyses of reliability and sharpness: the mean residual to address forecast bias, the residual sum of squares to address forecast accuracy, and the standard deviation of forecast probabilities to measure forecast sharpness (lines 259-280). These metrics are now included in the manuscript in Table 1.
3. A similar description of Scenarios A & B, their respective calibration curves and statistics, and a comparison of specific metrics between scenarios (lines 268-280).
4. An expanded description of the latitudinal trends in the spatially distributed sensitivity indices (lines 286-289).

In section 3.6, we added sentences to describe the statistics presented in Table 1, i.e., the forecast bias, accuracy, and sharpness. On lines 220-221, we added:

"Forecast reliability was assessed with the mean residual, a measure of bias, and the residual sum of squares, a measure of accuracy, between the $(\bar{p}_k, y_k)$."

On lines 225-227, we added a sentence to explain the connection between forecast sharpness and the standard deviation of forecast probabilities:

> "We used the standard deviation of forecast probabilities, $\sigma_p$, as a measure of forecast sharpness because a larger standard deviation indicates greater dispersion toward the extreme values (Bradley et al., 2019)."

On lines 293-297, we added content to a sentence to further describe uncertainty in debris-flow volume predictions. It now reads:

> "Even in the region for which it was developed, the prediction uncertainty associated with the EAV model can exceed an order of magnitude (Gartner et al., 2014; Barnhart et al., 2021), with greater uncertainties likely when models are applied in landscapes that differ from where they were trained (e.g., Gorr et al., 2023; Rengers et al., 2023). In the absence of sufficient data to train local or regional postfire debris-flow volume models, it may be possible to apply a correction factor to an existing model that was trained elsewhere (e.g., Rengers et al., 2023)."

On lines 325-329, we added the following sentences to highlight the ongoing difficulty in forecasting short duration, high intensity rainfall, which we believe is responsible for the under-forecasting bias in the WRF ensemble forecast simulation:

> "We expect that the large spread in peak $I_{15}$ in the WRF atmospheric model forecast and the uncertainties associated with it are not unique to our study area or the modeled rainstorm and should be considered in future applications of this and similar probabilistic debris-flow inundation model frameworks. Uncertainties in mesoscale precipitation forecasts of short duration, high intensity rainfall (e.g., peak $I_{15}$) are well documented even at lead times shorter than 24 hours (Cannon et al., 2020; English et al., 2021)."

**Additional changes to the manuscript**

In the Abstract on lines 22-24, we modified a sentence to clarify our results. It now reads:

> "Sensitivity analyses indicate that debris-flow volume and two parameters associated with debris-flow mobility exert significant influence on inundation predictions. However, reducing uncertainty in postfire debris-flow volume predictions will have the largest impact on reducing inundation outcome uncertainty."

On lines 101-102 and 176, we updated our use of units to be consistent with the NHESS submission guidelines, changing "mm/hr" to "mm h$^{-1}$".

On lines 104-105, we fixed a typo in the observed volume to be the correct value (679,000 m$^3$) and added a reference to the observed inundated area (more than 2,600,000 m$^2$).

On lines 118 and 299, we removed unnecessary hyphens.

In Section 3.2 on lines 122-123, we updated the first sentence to clarify that the WRF atmospheric modeling was completed in a different study. It now reads:

> "The 24-hour lead time, 100-member ensemble rainfall forecast for the 9 January 2018 event (Oakley et al., 2023) was generated using the Weather Research and Forecast (WRF) atmospheric model Version 4.3 (Skamarock et al., 2021)."

In several places, we changed instances of "-1", "0", and "1" to "negative one", "zero", and "one", respectively, in order to be consistent with the NHESS submission guidelines. These are on lines 154, 168, 186, 198, 224, 234, 264, and 265.

In Section 3.5 on line 164, we replaced "initiation points" with "basins" for clarification.

On line 304, we added a reference to Nyman et al. (2015) as an additional alternative methodology for predicting postfire debris-flow volumes.

On line 323, we fixed a typo by replacing "…nearly 60%..." with "…approximately 70%..."

On line 324, we replaced "precipitation" with "rainfall" for precision.

On line 330, we replaced a colon with a semi-colon and inserted "I$_{15}$" for clarity.

On lines 333-334, we supplemented a sentence for clarity with the phrase "…while threshold probabilities of 50% and 84% resulted in substantial underestimates of area inundated."

On line 337, we replaced "rainfall intensities" with "I$_{15}$" for clarity.

On lines 339-341, we split a sentence for clarity. It now reads "When using the observed I$_{15}$ to compute debris-flow volume (Scenario B), volumes were larger than what was observed. As a result, the model over-forecasted inundation and the calibration curve dropped below the diagonal one-to-one line (Fig. 5d)."

On line 364, we added a reference to Wells et al. (2023) to highlight another study that predicts burn severity in advance of fire.

We updated the References section to include references matching the new in-text citations. These are on lines 418-420, 423-425, 443-445, 472-474, 541-542, 559-565, and 603-605.

We updated the color schemes used in Fig. 1, Fig. 3, and the new Fig. 5 to be accessible to those with color vision deficiencies.

We updated the captions of Fig. 3 (lines 626-633), Fig. 4 (lines 636-639), Fig. 5 (lines 641-645), and Fig. 6 (lines 649-653).

References

Barnhart, K. R., Romero, V. Y., and Clifford, K. R.: User needs assessment for postfire debris-flow inundation hazard products, U.S. Geological Survey Open-File Rep. 2023–1025, 25 pp., https://doi.org/10.3133/ofr20231025, 2023.

Bradley, A. A., Demargne, J., and Franz, K. J.: Attributes of Forecast Quality, in: Handbook of Hydrometeorological Ensemble Forecasting, edited by: Duan, Q., Pappenberger, F., Wood, A., Cloke, H., Schaake, J., Springer, Berlin, Heidelberg, Germany, 849–892, https://doi.org/10.1007/978-3-642-39925-1_2, 2019.

Cannon, F., Oakley, N. S., Hecht, C. W., Michaelis, A., Cordeira, J. M., Kawzenuk, B., Demirdjian, R., Weihs, R., Fish, M. A., Wilson, A. M., and Ralph, F. M., Observations and predictability of a high-impact narrow cold-frontal rainband over Southern California on 2 February 2019, Weather Forecast., 35(5), 2083–2097, https://doi.org/10.1175/WAF-D-20-0012.1, 2020.

English, J. M., Turner, D. D., Alcott, T. I., Moninger, W. R., Bytheway, J. L., Cifelli, R., and Marquis, M., Evaluating operational and experimental HRRR model forecasts of atmospheric river events in California, Weather Forecast., 36(6), 1925–1944, https://doi.org/10.1175/WAF-D-21-0081.1, 2021.

Gorr, A. N., McGuire, L. A., Youberg, A. M., and Rengers, F. K.: A progressive flow-routing model for rapid assessment of debris-flow inundation, Landslides, 19, 2055–2073, https://doi.org/10.1007/s10346-022-01890-y, 2022.

Gourley, J. J., Vergara, H., Arthur, A., Clark III, R. A., Staley, D., Fulton, J., Hempel, L., Goodrich, D. C., Rowden, K., and Robichaud, P. R.: Predicting the Floods that Follow the Flames, B. Am. Meteorol. Soc., 101(7), E1101–E1106, https://doi.org/10.1175/BAMS-D-20-0040.1, 2020.

NOAA–USGS Debris Flow Task Force: NOAA-USGS debris-flow warning system—Final report, U.S. Geological Survey Circular 1283, 47 pp., https://doi.org/10.3133/cir1283, 2005.

Nyman, P., Smith, H. G., Sherwin, C. B., Langhans, C., Lane, P. N. and Sheridan, G. J.: Predicting sediment delivery from debris flows after wildfire, Geomorphology, 250(1), 173–186, https://doi.org/10.1016/j.geomorph.2015.08.023, 2015.

Rengers, F. K., Bower, S., Knapp, A., Kean, J. W., vonLembke, D. W., Thomas, M. A., Kostelnik, J., Barnhart, K. R., Bethel, M., Gartner, J. E., Hille, M., Staley, D. M., Anderson, J., Roberts, E. K., DeLong, S. B., Lane, B., Ridgway, P., and Murphy, B. P.: Evaluating Post-Wildfire Debris Flow Rainfall Thresholds

and Volume Models at the 2020 Grizzly Creek Fire in Glenwood Canyon, Colorado, USA, EGUsphere [preprint], https://doi.org/10.5194/egusphere-2023-2063, 2023.

Ripberger, J., Bell, A., Fox, A., Forney, A., Livingston, W., Gaddie, C., Silva, C., and Jenkins-Smith, H.: Communicating Probability Information in Weather Forecasts: Findings and Recommendations from a Living Systematic Review of the Research Literature, Weather Clim. Soc., 14(2), 481–498, https://doi.org/10.1175/WCAS-D-21-0034.1, 2022.

Wells, A. G., Hawbaker, T. J., Hiers, J. K., Kean, J., Loehman, R. A. and Steblein, P. F.: Predicting burn severity for integration with post-fire debris-flow hazard assessment: a case study from the Upper Colorado River Basin, USA, Int. J. Wildland Fire, 32(9), 1315–1331, https://doi.org/10.1071/WF22200, 2023.

Wilks, D. S.: Statistical Methods in the Atmospheric Sciences, Fourth Edition, Elsevier, Amsterdam, Netherlands, 818 pp., https://doi.org/10.1016/C2017-0-03921-6, 2019.

---

## Author Response (AR2)

In this response document, all line numbers refer to the newest draft of the manuscript (without track-changes).

Reviewer Report #1: Paul Santi

The model provides a rigorous and honest effort to predict inundation areas using existing models. As the authors state, this is an initial step to automate a process that will certainly become more important in the future.

My primary suggestion in this second review is in the order of presentation of findings in the abstract and conclusions. As written, the authors start with the fully predictive model which, unfortunately, has unconvincing results ("Approximately 94% of the observed inundated area was forecast to have an inundation probability greater than 1%"). Next, they indicate that the model improves when using actual volumes rather than predicted ones.

I think they paper would be stronger if the results are presented in the following order instead:

1. Predicted inundations using measured debris flow volume are strong

2. Predicted inundations using prediction models for volume are not strong

3. Part of under-forecasting can be attributed to I15 prediction

So, rather than starting with a "failed" model and basically digging yourself out of a hole to justify its use, you start showing that the concept is valid, but then start adding elements to show where the weaknesses are found.

> Response:
>
> We appreciate the reviewer's suggested reorganization of the Abstract and Conclusions and have made edits to implement this recommended structure. The relevant section of the Abstract (lines 14-24 in the newest manuscript draft) have been edited to read (edits in italics):
>
> "We applied this framework to simulate debris-flow inundation associated with the 9 January 2018 debris-flow event in Montecito, California, USA. *When the observed debris-flow volumes were used to drive the probabilistic forecast model, analysis of the simulated inundation probabilities demonstrates that the model is both reliable and sharp. In the fully predictive model, however, in which debris-flow likelihood and volume were computed from the atmospheric model ensemble's predictions of peak fifteen-minute rainfall intensity, $I_{15}$, the model generally under-forecasted the inundation area. The observed peak $I_{15}$ lies in the upper tail of the atmospheric model ensemble spread, thus a large fraction of ensemble members forecast lower $I_{15}$ than observed. Using these $I_{15}$ values as input to the inundation model resulted in lower than observed flow volumes which translated into under-forecasting of the inundation area. Even so, approximately* 94% of the observed inundated area was forecast to have an inundation probability greater than 1%, demonstrating that the observed extent of inundation was generally captured within the range of outcomes predicted by the model."
>
> The relevant section of the Conclusions (lines 380-386) was rearranged and now reads:
>
> "We applied this methodology by using a 24-hour, 100-member atmospheric model ensemble forecast of rainfall intensity associated with a destructive debris-flow event that followed the 2017 Thomas Fire. *When debris-flow volumes were well-constrained, the probabilistic model predictions were sharp and well-calibrated to the observed area inundated. In the fully predictive model*, approximately 99% of the observed inundation area was contained within a region where the

simulated probability of inundation was greater than zero. In general, however, we found that the model under-forecasted the area inundated. We attribute the under-forecasting of inundation extent to the fact that the observed peak 15-minute rainfall rates were in the upper tail of the atmospheric model ensemble distribution of forecast rainfall rates."

Also, in my reading, the influence of the volume prediction model was much more important than the accuracy of the I15 model, but the abstract seems to indicate the reverse. I think it would help to either change the emphasis, or if indeed the I15 model is more important, perhaps indicate this better in the body of the paper.

Response:

While the input debris-flow volume did exert the greatest control on the uncertainty associated with inundation outcomes, we believe that the model tendency to under- or over-forecast inundation was primarily controlled by the predicted value of peak $I_{15}$. This is clearest to see through comparison of the calibration curves of the fully predictive model, Scenario A, and Scenario B. We have added text to the Discussion section to make this point more explicit. Lines 336-345 in the newest manuscript draft now read (additions in italics):

"Our interpretation that the ensemble distribution of predicted $I_{15}$ led to under-forecasting is supported by *comparison with* the reliability diagrams associated with forecast scenarios A and B, which were run using observed debris-flow volumes and observed *peak* $I_{15}$, respectively (Fig. 5). The calibration curves from these two scenarios indicate high sensitivity of the calibration to the input debris-flow volumes*, which are influenced by peak $I_{15}$. When the volumes predicted from $I_{15}$ were too low, as in the fully predictive model, the calibration curve lies above the one-to-one line, indicating under-forecasting (Fig. 3b). The calibration curve passes through the one-to-one line when the observed volumes, which are greater than those computed using the ensemble predictions of $I_{15}$, were used (Scenario A; Fig. 5b). Finally, the calibration curve drops below the one-to-one line, indicating over-forecasting, when volumes are computed based on the observed $I_{15}$ (Scenario B; Fig. 5d and Fig. S1). Volumes computed from the observed $I_{15}$ were greater than both the observed volumes and those computed from the ensemble predictions of $I_{15}$. As a result, the model would have over-predicted inundation area if the atmospheric model yielded a perfect prediction of peak $I_{15}$.*"

Lines 16-21 - I appreciate the new information incorporated into the abstract, but there is some repetition and it could benefit from a rewrite to streamline the text.

Response:

We have removed repetitive information from the rewritten Abstract.

Lines 16-17 - Having a probability of inundation greater than 1% does not sound like a strong validation of the model to me.

Response:

We have clarified the intention of this statement in the rewritten Abstract by placing it at the end of the new sentence structure suggested by the reviewer (lines 22-24). The purpose of this statement

is to convey that even though the observed $I_{15}$ was on the extreme end of the atmospheric model ensemble, the inundation forecast was still able to capture the observed inundation patterns in the spread of model outcomes. This is a general objective of forecasting with model ensembles.

Key point in line 273 - when using actual volume, the model accuracy was much better

The authors have made a strong effort to address the reviewers comments, and in my opinion, they have sufficiently responded to justify publication, provided the modifications I suggest herein are incorporated as best they can.

Response:

We appreciate the reviewer's kind words and thoughtful critiques that have improved the quality of the manuscript.

I am sorry I do not think the manuscript has been completed. In the manuscript, only three pages are occupied by the parts Results and Discussion except related figures and tables. In addition, what are the innovative results obtained from this study? If the most important result is "a first step toward a near-real time hazard assessment product (line 25)", you have to give an in-depth discussion to verify the innovativeness of result.

Response:

To further support the line quoted in the reviewer's final sentence, we have added the following sentences (in italics) to the Discussion section (lines 356-365):

"The methods presented here take a step toward near-real time assessments of postfire debris-flow hazards associated with an incoming rainstorm. *Our work builds on that of Oakley et al. (2023), who used the same atmospheric model ensemble to produce probabilistic predictions of debris-flow likelihood and volume in watersheds burned by the 2017 Thomas Fire. They did not include predictions of postfire debris-flow inundation, but they identified that a product linking together postfire debris-flow volume ensembles with runout models was an important area of focus for future research to support impact-based decision making (Oakley et al., 2023). Further, recent surveys demonstrate a need for hazard assessment products that connect debris-flow inundation models with forecasts of rainfall in the short period of time between fire containment and the first precipitation event (Barnhart et al., 2023; Gourley et al., 2020).* Considering that decision quality improves when probabilistic information is presented appropriately in weather forecasts (Ripberger et al., 2022), the types of maps generated by the model framework presented here could be used to support decisions regarding evacuations, staging of equipment and emergency personnel, and debris-flow mitigation efforts."

Additional miscellaneous edits

Line 25: we combined sentences and replaced "However" with "but" for sentence fluidity.

Lines 366 and 372: we combined existing sentences with related content into one paragraph.